# Inpainting in Discrete Sobolev Spaces: Structural Information for Uncertainty Reduction

Marco Seracini [1,*,†] and Stephen R. Brown [2,3,*,†]

1. Department of Physics and Astronomy "Augusto Righi", University of Bologna, 6/2, Viale Carlo Berti Pichat, 40127 Bologna, Italy
2. Massachusetts Institute of Technology, 77 Massachusetts Ave., Cambridge, MA 02139, USA
3. Aprovechar Lab L3C, Montpelier, VT 05602, USA
* Correspondence: marco.seracini2@unibo.it (M.S.); srbrown@mit.edu (S.R.B.)
‡ These authors contributed equally to this work.

**Abstract:** In this article, we introduce a new mathematical functional whose minimization determines the quality of the solution for the exemplar-based inpainting-by-patch problem. The new functional expression includes finite difference terms in a similar fashion to what happens in the theoretical Sobolev spaces: its use reduces the uncertainty in the choice of the most suitable values for each point to inpaint. Moreover, we introduce a probabilistic model by which we prove that the usual principal directions, generally employed for continuous problems, are not enough to achieve consistent reconstructions in the discrete inpainting asset. Finally, we formalize a new priority index and new rules for its dynamic update. The quality of the reconstructions, achieved using a reduced neighborhood size of more than 95% with respect to the current state-of-the-art algorithms based on the same inpainting approach, further provides the experimental validation of the method.

**Keywords:** texture synthesis; inpainting by patch; image processing; discrete sobolev spaces





## 1. Introduction

In the literature, the so called "inpainting problem" consists of filling zones of "missing" information in an image, i.e., in a two-dimensional discrete set [1].

Focusing on the specific two dimensional image case, to have satisfactory results, the inpainted area has to be not "eye-distinguishable" from the rest of the given image by a human observer. This subjective way to evaluate the achieved results does not take into account the ground truth, in all of those cases when the to-be-inpainted areas are created deleting zones of the complete starting image (Even if in practical cases ground truth images are never available, it is possible to create specific tests where the area to reconstruct is expected to be exactly the part that has been deleted from the pristine image. This kind of test is useful, e.g., to verify the correct behavior of the reconstruction algorithms).

To tackle the inpainting problem different solutions have been proposed. On one hand, partial differential equations (PDE)-based approaches have been used. These methods involve the introduction of derivatives in the mathematical models (or their equivalent finite differences in the discrete case), working in the so called Sobolev spaces [2–7], i.e., in mathematical spaces where the norm of the Lebesgue spaces is extended to the derivatives of the function itself. PDE-based approaches achieve good results when the damaged regions are small in size; otherwise, their reconstructions appear blurred. Moreover, these methods usually need a considerable execution time [8–11].

On the other hand, an interesting promising technique was introduced by [12,13]: for each missing point in the to-be-inpainted area, a suitable neighborhood of the point is scrolled all over the known image data and the correct value is chosen maximizing a similarity measure between the neighborhood and the known signal (see, e.g., [12–20]). The quality of the reconstructions by this method is conditioned by the choice of an

appropriate similarity measure. Furthermore, despite its conceptual simplicity, this method provides very good results in a relatively affordable execution time [21–24]. The inpainting procedures based on this idea are referred as "inpainting-by-patch" or exemplar-based techniques, see Figure 1.

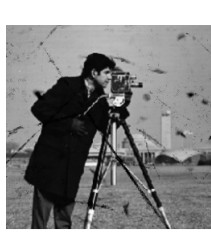 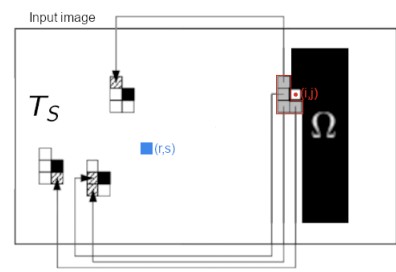 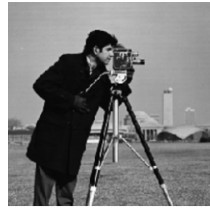

**Figure 1.** An example of a damaged image (left) restored (right) using the inpaint-by-patch approach (central scheme). The missing points in $\Omega$ (in black) are sequentially scrolled to individuate the best correspondence in the undamaged areas of the given image ($T_S$ in the scheme, in white). The values found are then used to fill the missing points.

In [15], a mathematical formalization describing the behavior of the algorithm introduced in [12,13] has been provided through a functional, named "Inpainting Energy" ($E_I$). The logic behind $E_I$ is to quantify the difference between a neighborhood of the missing points (reconstructed one by one) and the known part of the signal itself. Another even earlier work (see [25]), following in some sense a similar approach, used the correlation function in place of $E_I$: here the problem was the rigid registration of two similar images and the goal was the maximization of their similarity. However, we will show in Section 3 that the "deterministic" $E_I$ hides a strong probabilistic character: it is equivalent to finding, in the given image, which point has the highest probability to match the chosen neighborhood, considering a content-driven, patch-based similarity.

Both PDE-based methods and exemplar-based ones have the common goal, hidden by distinct mathematical formulations, to reduce the uncertainty in the choice of the missing points. In the first case, boundary conditions together with constraints on the structure (From here on, when we use the word *structure*, we will refer to the *shape*) of the three dimensional structure of the image function (in effect a surface immersed in a 3D space) and not to the geometry of the isophotes, as is usual in the image inpainting contest. of the unknown are needed (e.g., in [3,26], the discrete Laplacian is employed as a smoothness estimator); in the second case, the choice is driven by the best match among the available patches in the dataset, usually represented by the given part of the image (in the following sections we will refer to it as to the *training set* $T_S$, borrowing this expression from the neural networks literature).

Despite the goodness of the achieved results, inpainting-by-patch methods mask a conceptual weakness due to neglecting the under-the-surface structural information hidden in $T_S$. This structural information is connected with the concept of finite differences [27–29]. For this reason, introducing a similarity metric that cares about these structural aspects gives support to a better understanding of the logic behind the inpainting process.

Sobolev spaces [30], in their discrete versions, have been used in the image processing field with different formalization and aims (see, e.g., [31,32]).

We employ, in the context of image inpainting, the definition of Sobolev spaces to formulate a new, neighborhood dependent functional, described in Section 2: in our method we stay true to the patch-based idea, in a way that is quite different from classical PDE methods, in which higher order derivatives are employed for regularization reasons. To the best of our knowledge, this is the first time that a functional having this form is employed for the solution of the inpainting problem, such that it represents a novelty in the field of the inpainting-by.patch techniques.

More theoretically, in the seminal paper [29], a comparable approach for noise removal has been introduced in BV spaces, giving origin to a vast literature. Differently from [29], in our case, the finite differences are used as a similarity metric, as in [33] for textural classification purposes. An attempt in this direction has been described in [34], where a quite complete and general variational framework, applicable to super resolution and denoising problems, has been proposed. Like in [34], our results can be extended to other tasks, as image completion, texture synthesis, and image registration (only to cite some), thus providing general operational principles.

Another open debate for the solution of the inpainting problem, using the inpainting-by-patch methods, is focused on the scanning order of the missing points. The role of the scanning order is to inpaint first those points where the uncertainty is lower with respect to the others. The pivotal work exploiting the importance of the scanning order is represented by [35], in which a priority index has been introduced. The proposed procedure gave considerable results such that it is still commonly taken as reference for qualitative evaluations. For this reason, in what follows, we have compared our reconstructions to the ones achieved using [35]. Starting from [35], further attempts to improve the final quality have been performed by different authors (see, e.g., [36]), basing their works mostly on numerical considerations. Differently from them, we reformulate a completely new priority index, never used before in the literature, justifying its introduction by uncertainty reduction motivations.

An important scenario, for the inpainting problem, is represented by the fast and effective neural networks (NN) solutions. Recently, many inpainting methods have been developed based on deep learning techniques (see, e.g., [37–39]). The advantages of NN approaches reside in a fast computational time (after the time consuming initial training) with a high rate of success. Given these premises, the attempt to overpass the NN results appears an insuperable task. The only criticism one can make of NN is their well known lack of complete theoretical foundations, such that their design is essentially nowadays still driven by heuristics. An updated survey focused on NNs can be found in [40].

On one hand, the aim of our work is to investigate some hidden aspects of the deterministic inpainting procedure. On the other hand, this may guide the development of fast machine learning algorithms that have solid and understandable theory and provide benchmarks for comparison. Finally, we propose that some aspects considered in this work could be included in the formulation of new NN frameworks.

In what follows, the new functional formalization is given in Section 2, probabilistic considerations about the neighborhood are given in Section 3, the concept of causality is exploited in Section 4, and the importance of the uncertainty and scanning order is investigated in Section 5. The achieved results are discussed in Section 6. In Section 7 we show that, exploiting the knowledge of the relations between neighbor pixels, we can achieve qualitatively relevant reconstructions also operating a reduction in the training set $T_S$. Final remarks conclude the paper in Section 8. To improve the readability, in Appendix A, a list of symbols used and their meanings is provided. Technical numerical implementation details are described in the Appendix A.

## 2. Functional Formalization

Working with a W-bits coded image function $I$, of size $M \times N$ ( $M$ rows, $N$ columns), containing the area $\Omega$ to be inpainted (see Figure 2), we consider a function $\chi : [1, 2L+1]^2 \subset \mathbb{N}^2 \to [0, 2^W - 1]$ ($L \in \mathbb{N}$ determines the size of the square (Without loss of generality, the supp $\chi$ is assumed to be square. In the experimental part we will use also non-square-shaped neighborhoods.) supp $\chi$), that, centered in the generic point $(i, j)$ of $I$, is defined as

$$\chi_{i,j} := \chi(i + a, j + b) = I(i + a, j + b)$$

with $a = \{-L, -L+1, \ldots, L-1, L\}$, $b = \{-L, -L+1, \ldots, L-1, L\}$, $i \in [L+1, M-L]$ and $j \in [L+1, N-L]$ (We assume the first element of a matrix to be at coordinates (1,1)).

We will refer to supp $\chi$ in the following as the neighborhood of the point to be inpainted (see blue square in Figure 2 again). The choice as of a square-shaped support for $\chi$, with an odd value for its dimensions, simplifies the mathematical notation without loss of generality (On the contrary, this aspect must be taken into account in the numerical implementation). We will see in Section 3 how the shape and size of $\chi$ (i.e., value of $L$) is fundamental for the correct reconstruction of $\Omega$.

We outilne the inpainting task in three main sub problems:

1. Identifying the best *content-driven* solutions;
2. Identifying the best *structure-driven* solutions;
3. Inpainting using the best possible value, resulting from the combination of the best fits individuated in the two previous steps.

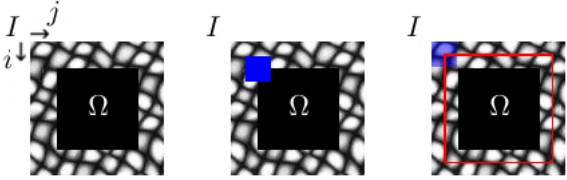

**Figure 2.** On the left the image $I$ contianing the to-be-inpainted area $\Omega$; in the center, the neighborhood $\chi$ in blue, centered in the top-left point to be inpainted; on the right, the red line represents the set $T_S$ of points for which $E_{C_T}$ is well defined.

In general, it is possible to define a pointwise distance function between two points of coordinates $(i,j)$ and $(h,m)$ in $I$ as

$$d(I(i,j), I(h,m)) := ||I(i,j) - I(h,m)||\,, \tag{1}$$

where $||\,.\,||$ is a generic norm. Using a similar approach as the one described in [15], the best *content-driven* correspondence for $\Omega$ is related to the minimization of the functional

$$E_{C_T}(r,s) := \sum_{(i,j) \in \Omega} E_C((i,j),(r,s)),$$

where, given a point of coordinates $(r,s) \in I \setminus \Omega$, $E_C((i,j),(r,s))$ is defined as

$$E_C((i,j),(r,s)) := \sum_{\substack{(p,q) \in \text{supp}\,\chi_{i,j} \setminus (i,j) \\ (h,m) \in \text{supp}\,\chi_{r,s} \setminus (r,s)}} d(I(p,q), I(h,m))\,. \tag{2}$$

The coordinates $(r,s)$ varies in a suitable subset $T_S \subset I \setminus \Omega$, such that the above sum is well defined without the introduction of further boundary conditions (e.g., red zone in Figure 2). In Equation (1), the $||\,.\,||_2$ norm, originally used in [15], has been replaced by a generic norm $||\,.\,||$. We will explicit this norm in the following.

To the light of the previous consideration, the functional $E_{C_T}$ represents a sort of *content-related* energy: it is the amount of error one commits comparing two patches having same support, equal to supp $\chi$.

In a fashion similar to the one used to formalize $E_{C_T}$, we introduce a new *structure-driven* functional, defined as follows

$$E_{S_T}((r,s),l) := \sum_{(i,j) \in \Omega} E_S((i,j),(r,s,),l),$$

where

$$E_S((i,j),(r,s),l) := \sum_{\Lambda} \sum_{k=1}^{l} \frac{1}{R_k} \sum_{\theta \in \Theta_k} d(\Delta_\theta^{(k)} I(p,q), \Delta_\theta^{(k)} I(h,k)),$$

$$\Lambda = \{p,q,h,m : (p,q) \in \text{supp}\,\chi_{i,j} \setminus (i,j), (h,m) \in \text{supp}\,\chi_{r,s} \setminus (r,s)\}.$$

In the above equation, $k$ is the order of the finite differences $\Delta_\theta^{(k)} I(p,q)$ and $\theta$ is their orientation, given the set $\Theta_k$ of all the available and *valid* orientations in the considered discrete set. A direction is considered *valid* if the calculation of the associated finite difference $\Delta_\theta^{(k)} I(p,q)$ does not involve the to-be-inpainted point of coordinates $(i,j)$. In the standard inpaint-by-patch process the value of $I$ in $(i,j)$ is excluded from the calculation of the pointwise differences: in the same way we excluded it in the finite differences calculation. In Figure 3 an example of the non-valid directions for the calculation of the finite differences of order one is given: the directions highlighted by the black arrows, connecting the red squares with the yellow ones are not valid, being $I(i,j)$ the unknown. The normalization coefficients are $R_k = 8k(\#\Theta_k - 1)$ with $\#\Theta_k = 8k$, where the symbol # denotes the cardinality of a set. The maximum order of finite differences $l$ is chosen accordingly with the size of $I$ and of $T_S$.

The finite differences in the direction $\theta = 0$, from order 1 to $k$, are defined as

$$\Delta_0^{(1)} I(p,q) = I(p,q+1) - I(p,q);$$
$$\Delta_0^{(2)} I(p,q) = \Delta_0^{(1)} \Delta_0^{(1)} I(p,q) = I(p,q+2) + I(p,q) - 2I(p,q+1);$$
$$\dots$$
$$\Delta_0^{(k)} I(p,q) = \underbrace{\Delta_0^{(1)} \dots \Delta_0^{(1)}}_{k \text{ times}} I(p,q).$$

**Figure 3.** Example of not valid directions for $k = 1$. The red square is the missing point to-be-inpainted, having coordinates $(i,j)$; the grid represents supp $\chi$ (being $3 \times 3$ in size for finite differences of order one); the arrows, connecting the yellow squares with the red ones, identify the non-valid directions: being centered in the yellow squares it is not possible to calculate the corresponding finite difference of order one because the value of the red square is unknown. It follows that, for each point, 7 directions are valid, and they are identified by the lines connecting the center of the white squares with the center of the red one. Indeed, in this example, $\Theta_1 = \{0°, 45°, 90°, 135°, 180°, 225°, 270°, 315°\}$, $\#\Theta_1 = 8$, $R_1 = 56$.

We recall that, being in a discrete case, the finite differences have a local (and not pointwise like the derivatives) character, as evident from the definitions above.

The logic paving the definition of $E_{S_T}$ is the same of $E_{C_T}$, except that the functional calculation involves, here, the finite differences for each available direction in the digital image set. For this reason, similarly to $E_{C_T}$, we interpret the functional $E_{S_T}$ as representative of a *structure-associated* energy, carrying information connected with the structure of the image content. The reason we employ the word "structure" stands from the fact that derivatives are representative of the shape, i.e., the "structure" of a function. Knowing the tangent to a function in each of its points is equivalent to knowing its structure, the same way the knowledge of an ordinary first order differential equation is effective for the knowledge of the unknown function, given the initial conditions.

The *content-related* term considers the pointwise differences only, while the *structure-associated* term takes into account of the local variation contribution.

Combining $E_{C_T}$ with $E_{S_T}$ in a new functional $E_T := E_{C_T} + E_{S_T}$, we can control the contributions related to both the content and the structure in the inpainting process. This combination represents a novelty in the exemplar-based inpainting-by-patch literature.

Being $E_C$ and $E_S$ dimensionally different, the introduction of normalization coefficients $\beta_k$, is needed. The expression of $E_S$ is, then

$$E_S((i,j),(r,s),l) := \sum_\Lambda \sum_{k=1}^l \frac{\beta_k}{R_k} \sum_{\theta \in \Theta_k} d(\Delta_\theta^{(k)} I(p,q), \Delta_\theta^{(k)} I(h,k)),$$

with $\beta_k \in \mathbb{R}$ (their values will be calculated in Section 3). If, and only if, $\chi$ is causal (According to the definition of causality in Signal Theory, a filter is defined causal if and only if the unknown value depends only on the ones given in the past. Strictly speaking, in the 2D asset, this means that only one single point, for each $\chi$, should be unknown (but this is never quite the concrete case). In Section 4 we will clarify this point.), then the minimization of $E_T$ gives the best choice for $I(i,j)$,

$$I(i,j) = I(\underset{(r,s) \in T_S}{\arg\min} \{E_T(r,s)\}),$$

where, as before, $T_S$ is a suitable set of coordinates such that, now, both $E_{C_T}$ and $E_{S_T}$ are well defined (For simplicity of notation we use, here, the same symbol $T_S$ as in Equation (2), even if, in general, the two sets could be not necessarily equal.). As we will show in Section 3, the introduction of $E_{S_T}$ reduces the uncertainty connected with the choice of the best possible value in $T_S$, for a missing point in $\Omega$.

Our formulation of $E_T$ models the inpainting problem such that we are able to take into account of the contribution of $E_S$, which has never been presented before in the existing reference literature (e.g., [35]).

## 3. Neighborhood

For simplicity of notation, in what follows we do not explicitly indicate the dependency of the energy from $(i,j)$ and $(r,s)$. The new functional $E_T = E_{C_T} + E_{S_T}$ hides a probabilistic interpretation. To prove this statement, we initially focus on $E_{C_T}$. For each point $(i,j) \in \Omega$, $E_C$ represents the sum of the differences, in modulus, in a domain of size supp $\chi_{i,j}$, between $\Omega$ and the subset $T_S \in I \setminus \Omega$: the closer this sum is to zero, the higher the probability is to find a correct match.

The value of $E_C$ is the sum of the $d((h,m),(p,q))$ contributions, as expressed in Equation (2). We can normalize each $d((h,m),(p,q))$ in supp $\chi_{i,j}$ as follows

$$d_{CNorm}((h,m),(p,q)) := \frac{d((h,m),(p,q))}{(2^W - 1)} \in [0,1]$$

such that we can define

$$E_{P_C}((h,m),(p,q)) := \begin{cases} d_{CNorm}((h,m),(p,q)), \ if \ d \neq 0, \\ \varepsilon, \ otherwise \end{cases}$$

where $\varepsilon$ is arbitrarily small and $E_{P_C}((h,m),(p,q))$ is the probability to have a match in supp $\chi_{i,j}$. Introducing a scanning order of supp $\chi_{i,j}$, e.g (without loss of generality) the raster scanning, we can write

$$E_{M_C}(i,j) := \prod_{\substack{(p,q) \in \text{supp} \chi_{i,j} \setminus (i,j) \\ (h,m) \in \text{supp} \chi_{r,s} \setminus (r,s)}} [1 - E_{P_C}((h,m),(p,q))]. \tag{3}$$

If $E_{M_C}(i,j)$ is close to one (ideally one for each term of the product) the reconstruction is exact, i.e., without errors of any kind. On the other hand, if $E_{M_C}(i,j)$ is close to zero, the probability to have a correct match is very low.

From Equation (3) two conditions follow:

- a hypothesis of statistical independence is hidden in the formulation;

- dropping a *ot equal to one* term in the product (3) is equivalent to drop part of the information related to the structure of $I$.

From the previous considerations it follows that the shape and the size of supp $\chi_{i,j}$ both determine the quality of the reconstructions. In particular, the second point states that it is not enough to take into account only the two orthogonal principal directions for the reconstruction (as it is usually desiderable in practice), because neglecting a point in supp $\chi$ is equivalent to assume to know its value with probability equal to one, i.e., that its value does not condition the outcome (size and shape of the neighborhood change the value of the *confidence* term introduced in [35]).

When the two above conditions, stating an "ideal" case, are not respected, the inpainting process does not generally perform correctly, the quality of the results depending on how far from the ideal case the real case is. We stress that supp $\chi$ is discriminating for content as it is for structure, but its shape (not necessarily square) and size are not sufficient for a correct reconstruction. In fact, if supp $\chi$ is made bigger and $E_{M_C}(i,j)$ stays close to one, we have a high probability of having made the correct choice. The other way around, if $E_{M_C}(i,j)$ decreases when supp $\chi$ grows, we have individuated a local similarity that is going to be lost increasing the "field of view" (i.e., the supp $\chi$). From the previous consideration comes that the correct choice of supp $\chi$ depends on the image content.

For what concerns $E_S$, in a similar manner used for $E_C$, it is possible to write

$$d_{SNorm}(\Delta_\theta^{(k)} I(p,q), \Delta_\theta^{(k)} I(h,m)) := \frac{d(\Delta_\theta^{(k)} I(p,q), \Delta_\theta^{(k)} I(h,m))}{2^k (2^W - 1)}$$

and

$$E_{P_S}((h,m),(p,q)) := \begin{cases} d_{SNorm}(\Delta_\theta^{(k)} I(p,q), \Delta_\theta^{(k)} I(h,m)), & if\ d \neq 0, \\ \varepsilon, otherwise \end{cases}$$

where, again, $\varepsilon$ is arbitrarily small and $E_{P_S}((h,m),(p,q))$ is the probability to have a match in supp $\chi_{i,j}$. Reasoning the same way we did for $E_C$, the *structural* patch matching probability $E_{M_S}$ can be expressed as

$$E_{M_S}(i,j) = \prod_{\substack{(p,q) \in \text{supp}\,\chi_{i,j} \setminus (i,j) \\ (h,m) \in \text{supp}\,\chi_{r,s} \setminus (r,s)}} [1 - E_{P_S}((h,m),(p,q))] .$$

Due to the normalization coefficient $2^k(2^W - 1)$, the contribution of the finite differences tends to "refine" the value of the inpainting energy by a factor $\beta_k \propto 1/2^k$. If we assume working with a bounded function, the hypothesis is always verified in the applications, and the values of $\beta_k$ scale exponentially with base 2, as evident from Figure 4. The example in figure describes an "extreme" case for which the contributions of subsequent finite differences stay fixed to one. When this does not happen, they tend to decrease as the order of derivation grows. We put in evidence that the determination of the normalizing factors makes our method non-parameter-dependent.

The structural information is hidden as in $E_{C_T}$ as in $E_{S_T}$. This consideration enlightens a subtle connection between content ($E_{C_T}$) and structure ($E_{S_T}$), being the structural information contained as in the size and shape of supp $\chi$ (i.e., in both $E_{C_T}$ and $E_{S_T}$), as well as in the finite differences expressions.

An application example is given in Figure 5 . Once the shape of supp $\chi$ has been chosen, the minimization of $E_T$ is equivalent to find the best match between patches: if supp $\chi$ is too small in size, it can occur to find considerably different values associated with the same $E_T$. In substance, the metric used for the calculation of $E_T$ gives origin to a set $S_C$ of equivalence classes, depending on supp $\chi$. The number of elements in each class decreases as supp $\chi$ becomes bigger in size as well as including $E_{S_T}$. In Figure 5, the effect of the introduction of $E_{S_T}$ on the trend of the trustability $C$ is shown: thanks to it, the

reduction in uncertainty $U = 1 - C$, given the same supp $\chi$ size, is faster. The inclusion of the term $E_{S_T}$ decreases $U$ as well as the space of the possible solutions. The role of $E_{S_T}$ can be seen as a refinement to discriminate between patches that would otherwise be equally eligible for the inpainting.

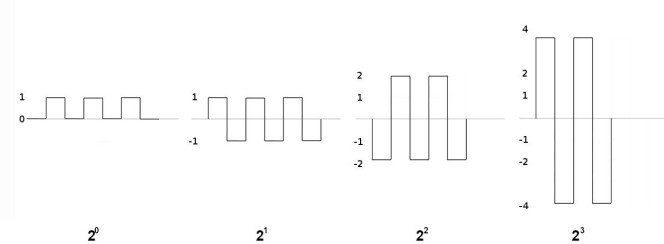

**Figure 4.** A square wave signal, with values in the range $[0, 1]$, represents the normalized signal with the highest amount of variation, whose maximum value is equal to $1 (= 2^0)$ (first figure on the left). We assume the values 0 and 1 to be respectively the minimum and the full scale of the measurement instrument. Performing the calculation of the finite difference of first order (second figure from the left) the range of variation expands to $2 = 2^1$ and so on, proceeding exponentially with the differentiation (last two figures on the right, relative to the second and third order finite differences, respectively).

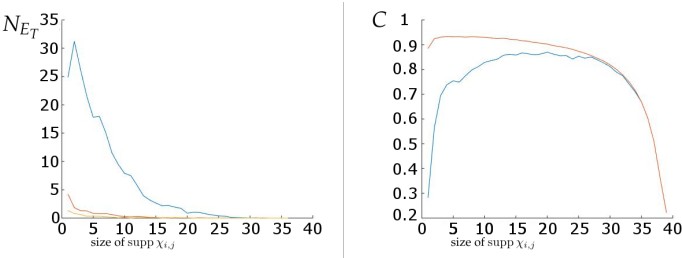

**Figure 5.** On the left, the trend of the number $N_{E_T}$ of patches having the same minimum value for $E_T$ when varying the size (side in pixels) of supp $\chi$; on the right the trustability $C (= 1 - U$, where $U$ is the uncertainty) in the choice of the available patches when varying supp $\chi$. The blue line is relative to the use of $E_{C_T}$ only, the red line gives reason for the introduction of $E_{S_T}$ (first order differences term in red, both first and second order difference terms in yellow). In particular, the closer $C$ is to one, the better the choice of the value for the reconstruction tends to be; on the contrary, the lower the number of candidate points is, the better the discrimination between different patches will be. In the right figure, $C$ (blue line) increases with the size of supp $\chi$ while tending to decrease when the number of available patches in the image becomes too small (right side of the graph on the right).

## 4. Causality

In signal theory, a system is defined to be *causal* if and only if the values of its output depend only on the present and past values of the input [41]. In the inpainting case, we state that the minimization of $E_T$ is meaningful if and only if supp $\chi$ guarantees the causality condition, i.e., if the calculation of $E_T$ is performed including only known points. If this does not happen, the values calculated, let us say in $(i, j + 1)$ in Figure 6, modifies the previously calculated ones $((i, j)$ in the same figure), whose number and position both depend on the size and shape of supp $\chi$.

The inpainting energy $E_T$ changes with time (the points of $\Omega$ are inpainted sequentially), such that the presence of unknown values for its calculation introduces loops in the system (feedback), making the minimization effectively not computable. Moreover, we remark that, in general

$$\min_{(r,s) \in T_S} \sum_{(i,j) \in \Omega} E_C((i,j), (r,s)) \neq \sum_{(i,j) \in \Omega} \min_{(r,s) \in T_S} E_C((i,j), (r,s)).$$

To the best of our knowledge, the previous inequality has been completely ignored by the existing exemplar-based inpainting-by-patch methods, in which the minimization is operated tacitly turning it to an equality (see, e.g., [15]). Another novelty of our work is, in fact, in the inclusion of this aspect of crucial importance in the inpainting dynamic.

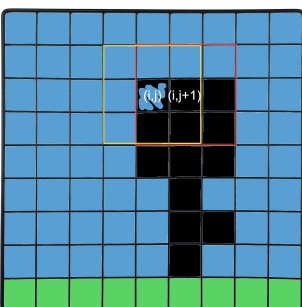

**Figure 6.** The inclusion of unknown points in $\operatorname{supp}\chi$, during the minimization of $E_T$, makes its calculation not effective. In the figure, it is assumed to use a square $\operatorname{supp}\chi$, highlighted in yellow when centered in $(i, j)$, i. e., at time $t$, and highlighted in red when centered in $(i, j + 1)$, i.e., at time $t + 1$. In fact, it results $\min\{E_T((i, j), t)\} \neq \min\{E_T((i, j), t + 1)\}$, where the min operation is performed all over the points belonging to the area that is not black.

The procedure described, e.g., in [13], is an example of an algorithm that does not take into account the causality and, to achieve convergence, sets random initial conditions in $\Omega$. Even if a convergence is reached, this way of proceeding has two main drawbacks:

- the inpainted result is not always the same, because it depends on the random initial conditions;
- if in $T_S$ the missing area is given, such that we expect to find a perfect match for $\Omega$ (i.e., $E_T = 0 \ \forall \ (i, j) \in \Omega$), the inpainting procedure does not generally reconstruct it as expected.

The missing of the causality condition is a source of uncertainty and its respect is needed to reduce the inpainting error. We take two actions to face the previous problems:

- we consider a variable $\operatorname{supp}\chi$ (in size and shape);
- we define an appropriate scanning order (priority) that takes into account at the same time of $E_T$, of the uncertainty connected with the shape of $\operatorname{supp}\chi$, and of the isophotes.

All the previous actions have been taken into account in our work, increasing its novelty character.

To be more specific, the first action is mainly algorithmic and consists in considering, in the calculation of $E_T$, only known points in $\Omega$ and in $T_S$ (A suitable binary mask is enough to numerically implement this condition. Thanks to it, whatever shape for $\operatorname{supp}\chi$ can be used and dynamically changed at runtime.). For this reason from now on, to simplify the notation, and due to the variability of $\operatorname{supp}\chi$, we will refer to the size of $\operatorname{supp}\chi$ to identify the minimum square size in which $\operatorname{supp}\chi$ is contained.

For the second action the formalization of a priority index is needed: being it a more complex procedure, we will discuss it in the following Section 5.

## 5. Uncertainty and Priority

The inpainting process consists of two different phases: the *analysis* of the similarity between patches (to determine the best fit) and the *synthesis* of the unknown value. Both of these processes hide sources of uncertainty.

In the analysis phase, the scanning order to inpaint $\Omega$ plays a central role for the correctness of the reconstruction. In this direction, different works have tried to improve

the confidence term firstly introduced in [35] (see [36,42,43] ). We shortly recall that in [35], the scanning order $P(i,j)$ (also called priority) has been defined as

$$P(i,j) = C(i,j) \cdot \frac{D(i,j)}{\alpha}$$

where $(i,j) \in \Omega$, $C(i,j)$ and $D(i,j)$ are, respectively, the confidence term (what we called trustability before), and the data term: the first one estimates the uncertainty of the neighborhood, the second one considers the direction of intersection between the isophotes and the boundary $\partial\Omega$ of $\Omega$. Briefly,

$$C(i,j) = \frac{\sum\limits_{(r,s) \in \text{supp}\,\chi \cap \text{supp}\,I} C(r,s)}{\#(\text{supp}\,\chi_{i,j})}, \quad D(i,j) = \frac{|\nabla^{\perp}_{(i,j)} \cdot n_{(i,j)}|}{\alpha} \tag{4}$$

where $\#(\text{supp}\,\chi_{i,j})$ is the cardinality of the pixels in supp $\chi_{i,j}$, $\nabla^{\perp}_{(i,j)}$ is the isophote vector and $n_{(i,j)}$ is the unitary vector orthogonal to $\partial\Omega$. $C(i,j)$ is initially set to one in every point of $I \setminus \Omega$ (see [35] for a complete description). The value of $P(i,j)$, being in the discrete, and consequently approximated asset, is strictly connected with the methods used to achieve $\nabla^{\perp}_{(i,j)}$ and $n_{(i,j)}$, whose calculation usually requires the introduction of a preprocessing stage by a Gaussian smoothing to reduce noise. Dramatically different reconstructions come out from different implementations of the same method.

The contributions of $C(i,j)$ and $D(i,j)$ have been usually considered separated, as respectively explanatory of the uncertainty of the given data and the preferential direction of propagation of the isophotes [42,43]. Both $C(i,j)$ and $D(i,j)$ belong to the interval $[0,1]$, given a suitable normalization coefficient $\alpha$ (e.g., 255 if W=8). Exploiting this property, we can interpret them as probabilities. More precisely, $C(i,j)$ is the probability to find a match, given a certain *trustable* number of neighbor points, while, when $D(i,j)$ is normalized, it expresses a probability connected with the structure, i.e., the morphology of $I$ around $\Omega$.

The last statement follows from the properties of edges in images. In fact, points belonging to contours of an object are those points that generally are less numerous than other points, i.e., that are less probable to be found.

An explanatory example is provided in Figure 7: to inpaint the Kanizsa [44] triangle we have to set the priority. As an example, we take into account of two plausible configurations of $\chi$, being supp $\chi$ highlighted in light green in Figure 7 and shown in the small squares on the right (in black the missing areas). It is evident that the number of points, in $I \setminus \Omega$, compatible with the upper square is lower than the number of points compatible with the lower one.

Thought in this way, $D(i,j)$ individuates the directions along which there is a reduction of uncertainty in the choice of the best fit. In fact, the equivalence class to which the candidate points belong, has (auspicably) the lowest cardinality with respect to the other ones in $S_C$. The probability to make a mistake, in the choice is, then, reduced even in presence of multiple candidates. For this reason, we introduce $D_M(i,j)$, in place of $D(i,j)$, defining it as

$$D_M(i,j) := \max_{\theta} \{ \Delta^{(1)}_{\theta} I(i,j) \} . \tag{5}$$

Edges are zones where exists a $\Delta^{(1)}_{\theta}$ that is higher than other areas of the image. Then, its maximum individuates the direction of lowest uncertainty.

Two main advantages follow from our new formalization:

- no need to calculate (and approximate) $\nabla^{\perp}$;
- there are not preferential straight directions of propagation, a known drawback of [35].

The reasons we limit the value of $D_M$ to the first order of the finite differences in the definition (5) have numerical and perceptual nature and are explained in the Appendix A.

Moreover, back to the probabilistic interpretation given to $E_T$ in Section 3, its value can be considered a measure of uncertainty itself: where $E_T$ is lower, the uncertainty is lower the same way. Then, we include the value of $E_T$ in the final formalization of our new priority index:

$$P^*(i,j) = [C(i,j) + a^* D_M(i,j)] \cdot b^* E_T(i,j)$$

where $a^*$ and $b^*$ are two normalization coefficients.

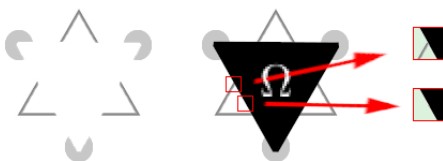

**Figure 7.** On the left, the Kanizsa triangle to inpaint; in the center, the same tringle with $\Omega$ highlighted in black; on the right, two examples of neighborhood to be inpainted in the black areas. It is easy to verify that the number of possible patches compatible with the top neighborhood is lower than the number of patches compatible with the down one.

The value of $a^*$ follows from two considerations: $D_M$ is a finite difference subject to the same normalization reasoning of Section 3, and it has to be normalized by the ratio of valid points involved in its calculation, i.e., $C(i,j)$. This last consideration has to be extended to $E_T$ too, providing the value of $b^*$. From these premises follows that, in the case of finite differences of order one (that we used in practical applications), it results $a^* = C(i,j)/255$ and $b^* = C(i,j)$.

The new $P^*$ balances between the "morphological" uncertainty connected with the shape and size of $\Omega$ and its surroundings (i.e., $[C(i,j) + a^* D_M(i,j)]$), and the uncertainty due to a limited $T_S$ (i.e., $b^* E_T$).

To quantitatively support our choice, we inpaint a triangular shape (see Figure 8) the same way they did in Figure 9 of [35]: in our case, it is possible to see that no "over-shot" artifacts appear. The completion of the triangle is not as expected, but this is due to the lack of the right patch in the rest of the known image: no top-pointing corners are available without rotation.

In Figure 9 the reconstruction by our algorithm of a single straight line shows the connectivity principle is respected.

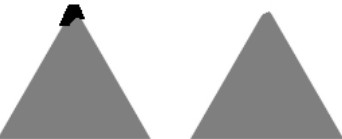

**Figure 8.** On the left, the triangle to be inpainted ($\Omega$ in black); on the right, the result of inpainting using $2L + 1 = 3$ and $l = 1$. No "over-shot" artifacts appear. Moreover, the result is quite symmetric, even if $\Omega$ is not. The expected top-pointing corner is not achieved because it misses in the known part of the image. The convergence to the stable solution is achieved at the 4th iteration.

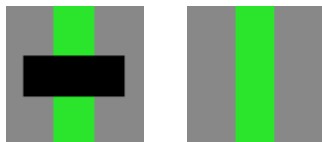

**Figure 9.** On the left the image to inpaint ($\Omega$ in black); on the right the inpainted result ($2L + 1 = 3$, $l = 1$). Our algorithm guarantees the connectivity principle, whatever is the size of $\Omega$ (result according with [34]).

Back to $P(i,j)$, another crucial point for its calculation is represented by the esteem of $C(i,j)$: in Equation (4), the known points in supp $\chi \cap$ supp $I$ are set to one, independently

from $\Omega$. In probabilistic terms, this hypothesis is equivalent to state that all the given points in supp $\chi \cap$ supp $I$ have the highest *trustability*, i.e., their values are certainly correct. This assumption is true if there is a perfect match between the points to inpaint in $\Omega$ and the available ones in $T_S$, but in the most of the real cases this does not happen. In fact, generally speaking, $E_T(i,j) \neq 0 \, \forall (i,j) \in \Omega$ (otherwise the solution of the inpainting problem would be obvious). Moreover, we remind that a reference image is unavailable for the quantification of the quality of the reconstructions.

From the previous considerations it follows that $E_T(i,j) \neq 0$ has an effect on $C(i,j)$, when $(i,j) \in I \setminus \Omega$, and this effect has to be propagated to the whole image, i.e., the error has to be redistributed. This reasoning is licit only because a reference is missing (otherwise $E_T(i,j) = 0 \, \forall (i,j)$, as noted before): only in this case, in fact, if we look at the image from the reference system of the missing points in $\Omega$, we can assume the surrounding points are wrong, viceversa if we look to the problem taking as reference the points in $I \setminus \Omega$. This considerations determine the need to upload $C(i,j)$ also in $I \setminus \Omega$, given that, by hypotheses, the value of $E_T(i,j)$ is not zero.

In Figure 10 a graphical example is provided.

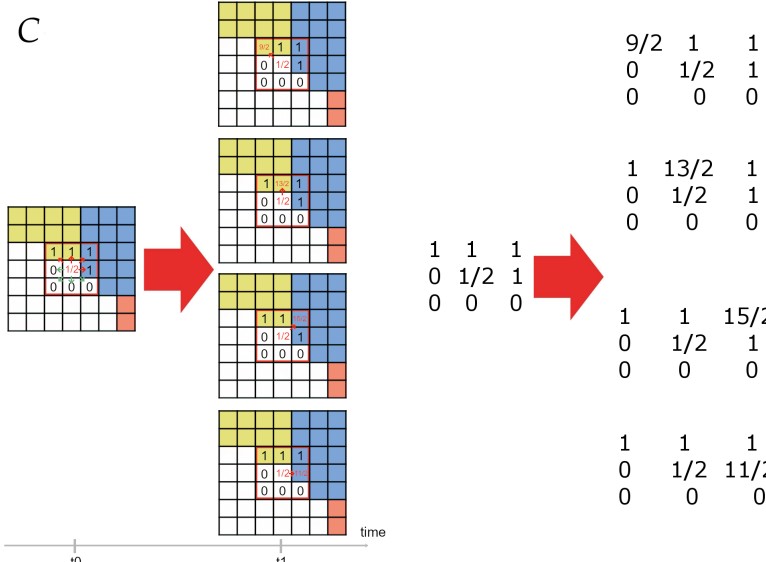

**Figure 10.** Description of the updating process: time arrow from left to right, $\Omega$ in white. On the left, at time $t_0$, the small green and red arrows around the point to inpaint (in the center of the red highlighted square, coordinates $(i,j)$, $C(i,j) = 1/2$) determine the directions for the updating of the trustability. On the right, at time $t_1$, the propagation of the new value (corresponding zoommed numbers on the right): while it is tacitly assumed fair to update the values of $C(i,j)$ in $\Omega$ (green arrows, starting zero value), the same has to be performed also in $I \setminus \Omega$ (red arrows, starting value equal to one). The linearity allows to update $C(i,j)$ individually for each single point and to sum the results. The Markovian hypothesis limits the radius of influence for the update to the bordering pixels only.

We can figure $C$ to be as a field (e.g., of force) where all the points are interconnected: a perturbation occurring somewhere would cause the modification of the entire configuration. To make this update numerically plausible we postulate two conjectures: we assume

- the system must satisfy the superposition effects principle (linearity);
- the system must be Markovian [45].

Both these conjectures are needed with the uinque scope to make the problem computable. In fact, the first one allows to update $C(i,j)$ in $I \setminus \Omega$ summing the effects separately; the second one limits the radius of influences of the perturbation to the bordering neighborhoods. If the linearity is missing, the computation would not arrive to an end, because

changing the value of one pixel will influence the minima connected with the other points in $\Omega$. If the the system is not Markovian, the chainging in the value of a missing point would propagate all over the other values, producing an infinite loop, as in the case of the lack of linearity.

To the best of our knowledge, this is the first time that a recalculation of the priority is performed redistributing the error. In Figure 11 we provide an example of the value of $P^*$ after the first iteration for the reconstruction given in Figure 12: the priority decreases from the borders of $\Omega$ to the center of it.

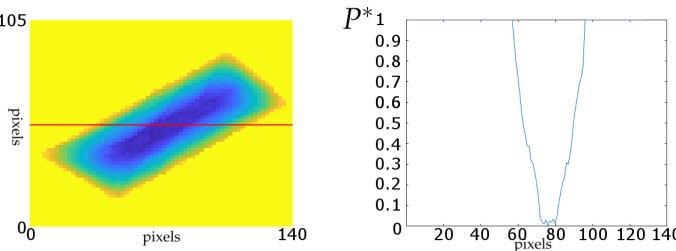

**Figure 11.** On the left: in false colors, the values of $P^*$ after the first iteration for the reconstruction of the image shown in Figure 12 (detail). Points in bright yellow have values of $P^* = 1$; the other colors denote values lower than one. On the right the trend of $P^*$ for the red line highlighted on the left picture. Image size of $140 \times 105$.

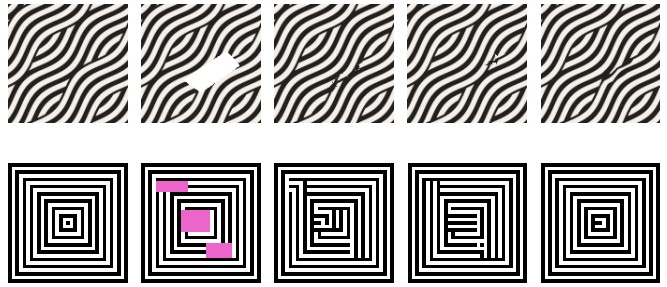

**Figure 12.** From left to right: original image, image to be inpainted ($\Omega$ in white (**top**) and pink (**down**)), reconstruction by [35], reconstruction by Matlab *inpaintExemplar* function, reconstruction using our method ($2L + 1 = 3, l = 1$). Our proposed method exhibits more regularity respect with the other two algorithms.

Neglecting our proposed updating process brings to visual poor results, as shown in Figure 13.

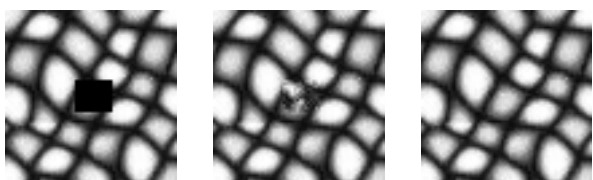

**Figure 13.** On the left, the texture to inpaint ($\Omega$ in black). In the center, the inpainting result achieved if $C(i,j)$ is not updated in $I \setminus \Omega$ (i.e., $C(i,j) = 1 \,\forall (i,j) \in I \setminus \Omega$, $T_S = I \setminus \Omega$). On the right, the inpainting result updating the points in $I \setminus \Omega$ ($2L + 1 = 3, l = 1$). The improvement of the result, when correctly updating $C(i,j)$, is evident.

For what concerns the synthesis phase, a source of uncertainty is due to the norm used to quantify the patch similarity. In past works it has been stressed how the most appropriate norm for images is the TV norm, that is, essentially an $\ell^1$ norm of derivatives [29]. On the other hand, in [34], results achieved using the $\ell^2$ norm are evaluated as "smoother" and, for this reason, better than the $\ell^1$-based ones. For this reason, and according with the most of the literature, we have chosen to operate with the $\ell^2$ norm.

### 6. Numerical Results

In this section we show numerical results achieved using $E_T$. No limitations are imposed on the topology of $\Omega$.

The first toy example, here discussed to recall and clarify the novelties introduced in the previous sections, inpaints one single point of coordinates $(i, j)$ in a periodic chessboard-like pattern, such as the red one in Figure 14. As result of the reconstruction, we expect a white value to fill the vacancy. According to what stated in the theoretical section, we assume $(i, j)$ to be possibly located in whatever position of supp $\chi$ and not necessarily in its center. We name the configurations arising from this choice from 3.1 to 3.9 and we group them in the set of patterns $S_C = \{3.1, 3.2, 3.3, 3.4, 3.5, 3.6, 3.7, 3.8, 3.9\}$. Each element of $S_C$ is marked by a couple of integers separated by a dot: the first number gives the size of one side of supp $\chi$, and the second one specifies the pattern position in the set. In fact, $S_C$ individuates the equivalence classes mentioned in Section 3.

The number and elements of this set are shown in Figure 15 for $k = 1$. Being that the red point is surrounded by known values, each configuration of the set $S_C$ is causal. The patches compatible with the configurations $S_C$ and giving a correct solution, taken from the chessboard of Figure 14, are shown in the right part of Figure 15.

With each configuration of $S_C$ is associated a set of points compatible with the reconstruction: for each element of $S_C$, they are shown using different colors in Figure 16: we name each set with $\mathcal{S}_{3.1}, \mathcal{S}_{3.2}, \ldots, \mathcal{S}_{3.9}$.

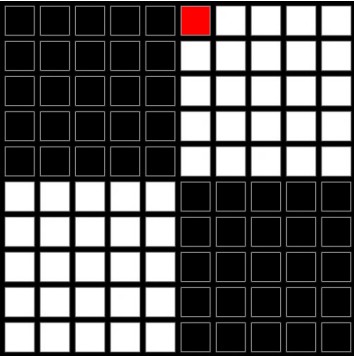

**Figure 14.** In red, the point to inpaint in the periodic chessboard-like pattern. Each square of the chessboard consists of $5 \times 5$ pixels.

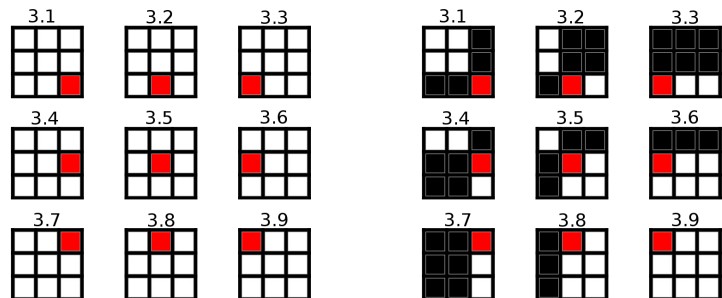

**Figure 15.** On the left: all the possible position of the missing point (in red) in supp $\chi$. On the right: each $3 \times 3$ sized pattern compatible with the missing point of Figure 14.

The exact solution, i.e., the one taking into account as the content (black or white color), as the structure (i.e., the fact that the missing point is on a corner) comes from sets $\mathcal{S}_{3.1}, \mathcal{S}_{3.2}, \mathcal{S}_{3.4}, \mathcal{S}_{3.5}$ only. The other ones correctly identify the white value but the result is achieved not taking into account the particular position of the red point in the chessboard. In more complex and general cases, when there is not a perfect match in $T_S$, the role played by $E_{S_T}$ increases in its importance, such that considering $E_T = E_{C_T} + E_{S_T}$ is crucial.

An example of the inpainting of a chessbord with our method is provided in Figure 17. Other numerical cases are considered in what follows.

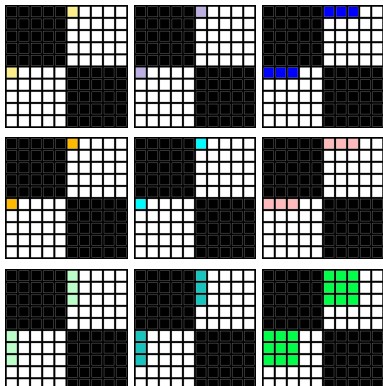

**Figure 16.** From top to down, left to right: the candidate points for each different pattern (from 3.1 to 3.9). The squares with the same color in the 9 chessboards constitute the sets $\mathcal{S}_{3.1}, \mathcal{S}_{3.2}, \ldots, \mathcal{S}_{3.9}$.

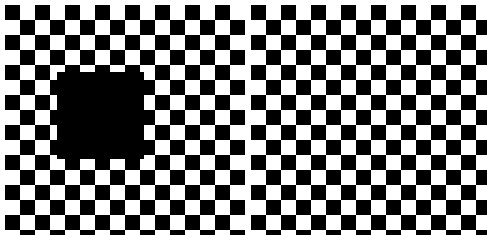

**Figure 17.** Inpainting of the missing part $\Omega$ in a chessboard, using $2L + 1 = 3$, $l = 1$.

We underline that, starting with non-random initial conditions (e.g., all zero values), state-of-the-art methods like [13] fail (see Figure 18). Indeed, the respect of the causality condition assures that the inpainting process is independent from the initial conditions.

In a quasi-periodic texture (de Bonet's sample number 161 [46]), $\Omega$ has size of $12 \times 14$ pixels (the black rectangle in Figure 19).

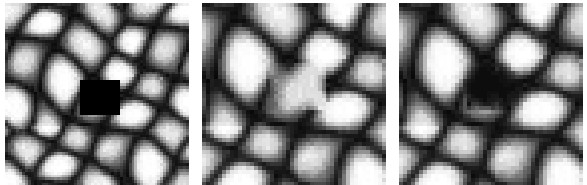

**Figure 18.** Inpainting of $\Omega$ using [13], for different values of supp $\chi$ and different initial conditions. On the left: initial condition with random white noise; on the right: initial condition equal to zero. Both images achieved using $2L + 1 = 3$. This is an example of missing causality condition.

Using the method proposed in [13], as the size of the neighborhood increases, the reconstruction quality improves (its minimum size for a correct inpainting is $2L + 1 = 13$). Using our formalization with $E_T = E_{C_T} + E_{S_T}$, in place of $E_T = E_{C_T}$, even the case $2L + 1 = 3$ results in a correct reconstruction (see Figure 19). With our methods, the needed number of surrounding points, for a correct inpainting, can be estimated to be $3^2/13^2 \approx 0.05$ (5%) of the amount of the ones needed in [13] to achieve results having a quality comparable with our reconstructions.

In [15], they formulated the expression of the inpainting energy and showed that it has a decreasing trend. We perform the same, achieving that our new version $E_T = E_{C_T} + E_{S_T}$ not only has a decreasing trend but reaches the minimum value faster (see Figure 20, that is qualitatively representative of all the performed tests).

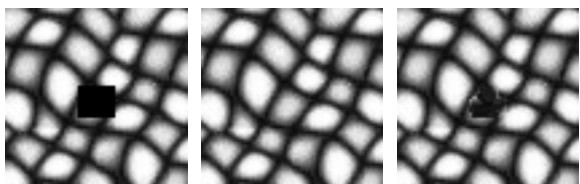

**Figure 19.** From left to right: initial $\Omega$ of size $12 \times 14$ pixels (**left**); inpainting result using our method ($2L + 1 = 3, l = 1$) (**center**); inpainting result using method [13] with $2L + 1 = 11$ (**right**). The better performance deriving from the introduction of $E_{S_T}$ and the causality of the new proposed method are visually evident. The image in the center is achieved at the first iteration and does not change meaningfully in the subsequent ones, highlighting a fast convergence too. Our method needs less than 8 points to provide suitable results, while for [13], $11^2 - 1 = 120$ points are not enough.

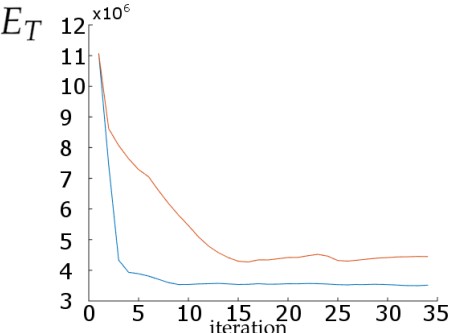

**Figure 20.** Figure shows $E_T$ trends, calculated at the end of each scanning iteration of the whole $\Omega$: in red $E_T = E_{C_T}$, in blue $E_T = E_{C_T} + E_{S_T}$. Finite difference contributions determine a faster convergence to the steady state.

**Remark 1.** *The value of $E_T$ is meaningful when it is calculated at the end of each scan and if the causality condition is respected. Otherwise, in the inpainting algorithm, a potentially dangerous (for stability) positive feedback is introduced. During the first scan, the causality property addresses the algorithm to the direction of that minimum characterized by the lowest uncertainty. Subsequent scans refine the calculation.*

The introduction of $E_{S_T}$ allows to carry more structural information in the similarity computation.

Other examples of reconstructions for quasi-periodic textures are provided in Figures 21 and 22.

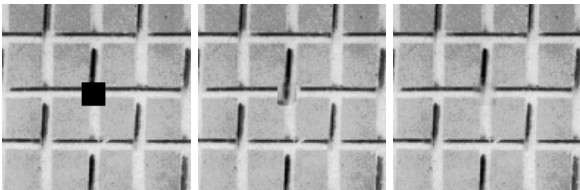

**Figure 21.** Inpainting of texture D1 taken from Brodatz database. From left to right: image to be inpainted ($\Omega$ in black, size $16 \times 16$); inpainting by mean of $E_T = E_{C_T}$ with $2L + 1 = 23$ (second column) and $E_T = E_{C_T} + E_{S_T}$ ($2L + 1 = 3, l = 1$) (third column). The case using the first order finite difference reconstructs visually better. Moreover the minimum value of $L$ to inpaint correctly is 23 when $E_T = E_{C_T}$, against $L = 3$ when introducing $E_{S_T}$ (reduction in the number of needed points of around 98%).

Moreover, we compared the proposed method with results achieved by Matlab© *inpaintExemplar* function ([35,47]) and with ones coming from an available implementation of [35] (see Figures 23–26). In Figure 24, the reconstruction of the Kanizsa triangle is shown: the connectivity principle is respected.

A video showing the dynamic of the inpainting process for different classes of images is available at https://youtu.be/GPGTA_Ukt9g (accessed on 16 August 2023).

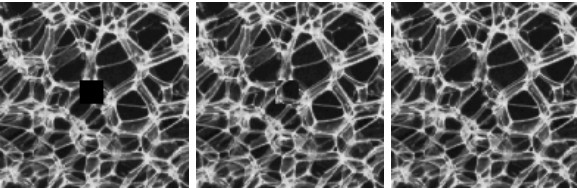

**Figure 22.** Inpainting of texture D111 taken from Brodatz database. From left to right: image to be inpainted ($\Omega$ in black, size $16 \times 16$); inpainting by mean of $E_T = E_{C_T}$ with $2L + 1 = 27$ (second column); inpainting by mean of $E_T = E_{C_T} + E_{S_T}$ with $2L + 1 = 3$, $l = 1$ (third column). The case using the first-order finite difference reconstructs visually better. Note, again, the different value of $L$ (reduction in the number of needed points of around 99%).

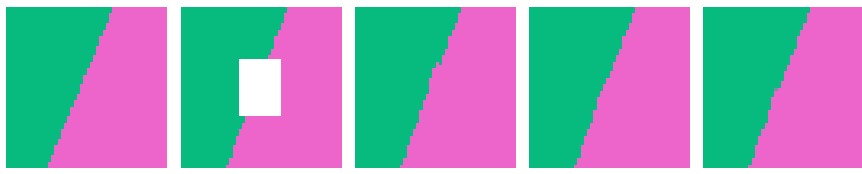

**Figure 23.** From left to right: original image, image to be inpainted ($\Omega$ in white), reconstruction by [35], reconstruction by Matlab *inpaintExemplar* function, reconstruction using our method ($2L + 1 = 3$, $l = 1$). All the methods guarantee the connection of the isophote along a straight line, according to the connectivity principle.

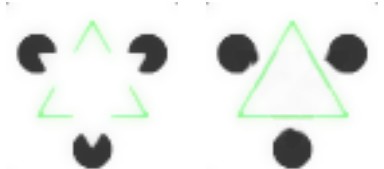

**Figure 24.** Inpainting of the Kanizsa triangle. The starting image (**left**); the inpainting result, achieved using $2L + 1 = 3$, $l = 1$ (**right**).

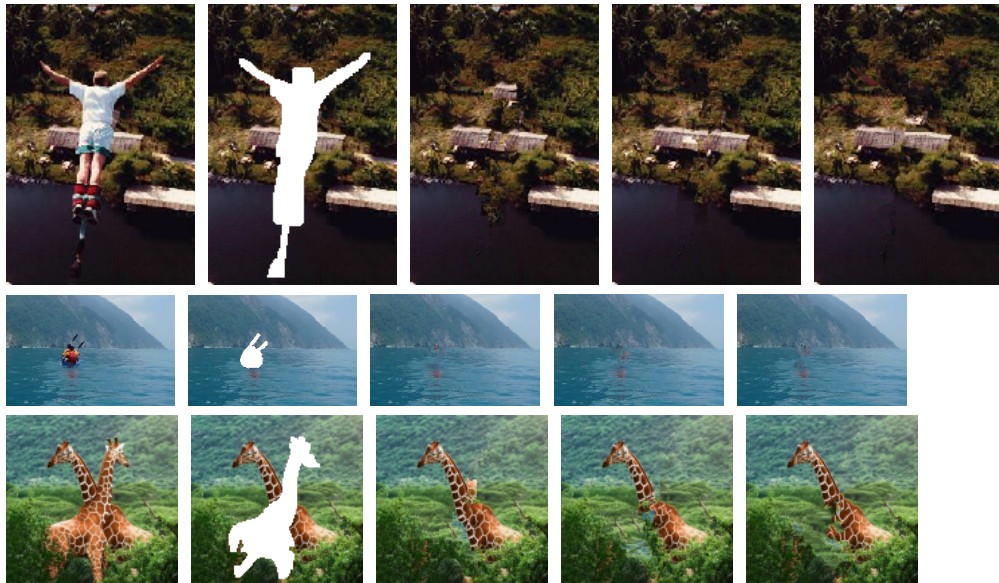

**Figure 25.** *Cont.*

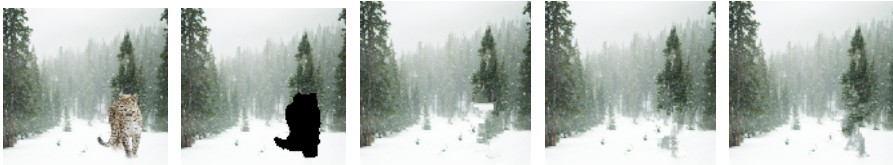

**Figure 25.** Reconstructions from different inpainting methods. For each image set, first row: original image (**left**); image to inpaint with Ω (**center**); reconstruction by [35] (**right**). Second row: Matlab *inpaintExemplar* reconstruction (**left**); reconstruction by our method ($2L + 1 = 3$, $l = 1$) (**right**).

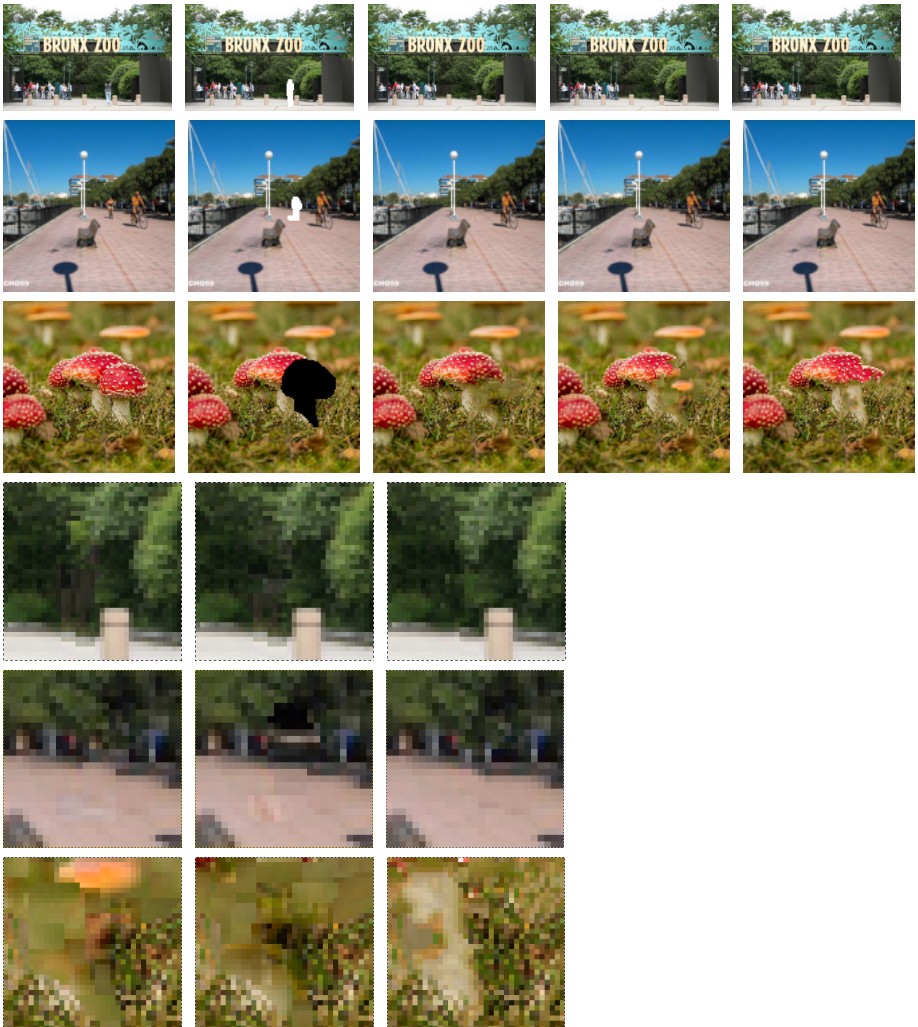

**Figure 26.** Reconstructions and their details. From top: original and inpainted image sets in the same order of Figure 25. In the last three rows, details of the achieved reconstructions: Matlab *inpaintExemplar* (**left**); [35] (**center**); our method (**right**). At this level of zoom, our method exhibits a better continuity in the structure of the inpainted zones.

## 7. Training Set

A pivotal role in the inpainting procedure is played by the training set $T_S$. In particular, as the size of $T_S$ decreases, the quality of the reconstructions degrades, due to the lower number of available patches. On the other hand, having a $T_S$ that is too big can result in wrong inpainted areas, especially in those points far enough from ∂Ω. Another drawback of a too-extended $T_S$ is the computational time.

Furthermore, the impressive results achieved in [48] using neural networks prove that the information provided in images are redundant, such that even a limited subset is enough to inpaint correctly the missing areas.

Moreover, advances in the field of texture synthesis [16] have been based on the same hypothesis, such that a reduced $T_S$ appears to be an effective policy to correctly inpaint.

In other works, it is stated that the inpainting process proceeds by continuation of the structure of the area surrounding $\Omega$ into it, such that contour lines are drawn via the prolongation of those arriving at $\partial\Omega$ [3]. Moreover, other techniques have based their effectiveness on metrics including the proximity between the point to inpaint and the match found in $T_S$, to penalize the ones at higher distance (see, e.g., [31,49]).

In the light of the previous observations, we can reduce $T_S$ to include only limited areas of $I \setminus \Omega$: some examples operating this reduction are available in Figure 27.

Other examples of a correct inpainting using a reduced training set are shown in Figure 28, using pictures from [50].

One more time, the results achieved reducing $T_S$ put in evidence how our inclusion of the finite differences in the inpainting procedure, reduces the number of points needed for a correct reconstruction exploiting the information hidden in the relations between neighborhoods.

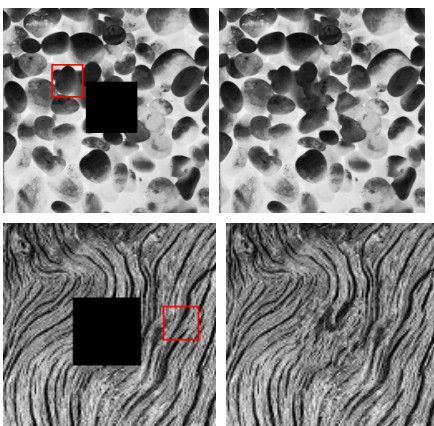

**Figure 27.** Inpainting of a stone and a wood structure. The inpainting results have been achieved with $2L + 1 = 3$, $l = 1$. The reduced training sets used to inpaint are contained in the red squares: their size is modest if compared with $I \setminus \Omega$.

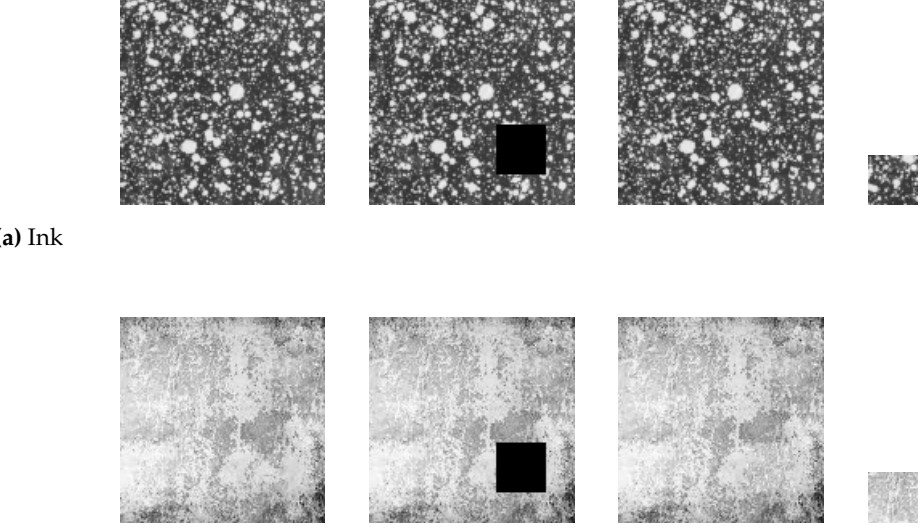

**(a)** Ink

**(b)** Marble

**Figure 28.** *Cont.*

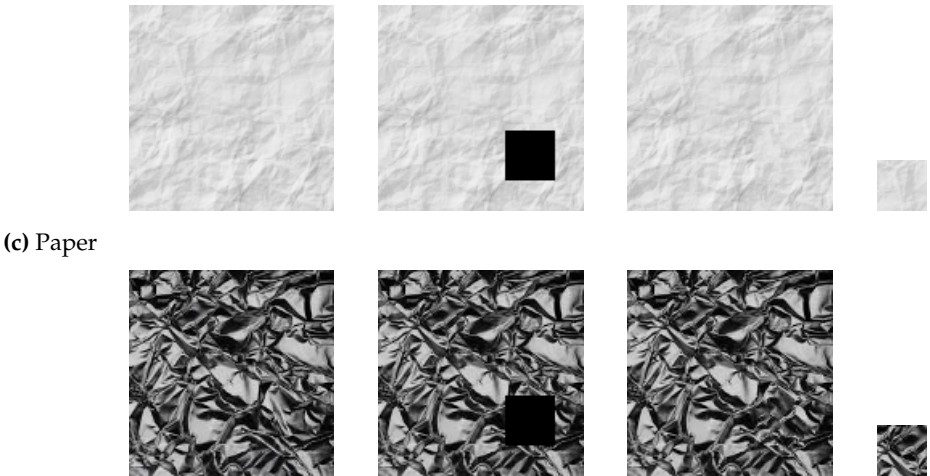

**(c)** Paper

**(d)** Metal

**Figure 28.** Inpainting of textures from [50] (a) ink; (b) marble; (c) paper; (d) metal. From left to right: the pristine image; the image to inpaint ($\Omega$ in black); the inpainted result; the used training set (same scale of the images).

## 8. Conclusions

In this paper, we have introduced a new functional to investigate the inpainting problem: it is strictly connected with the definition of Sobolev spaces but formalized in discrete setting. We have replaced the usual MSE-like metric (mean square error) with a more precise one, in the Sobolev spaces sense, for the determination of the correct values to inpaint.

The results we achieved show that the new functional, taking into account finite differences, provides reconstructions of good quality.

The improvement is, on one hand, theoretically predictable (functions being equal, or similar, for higher order derivatives "resemble" more each other), and, on the other hand we provided effective practical examples.

In particular, the new introduced functional formulation carries, in the reconstructed area, precise structural information, depending on the size and shape of the neighborhood, which strongly conditions the final quality of the reconstructions. An application proof of this behavior stands in the capability of the new method to reproduce and connect isophotes, even in complex cases, not necessarily privileging the propagation along straight lines.

Moreover, the new formulation plays two complementary roles: on one hand it speeds up the convergence toward a minimum value; on the other hand, it reduces the number of local minima, lowering the probability to end the process in one of them.

To explain the role of the neighborhood in the inpainting process, we have introduced a statistical model, thanks to which we have proved that considering only the principal directions is not sufficient to achieve correct reconstructions.

In addition, from our investigation of the role played by the causality condition, we have shown it represents a starting point in the direction of the uncertainty reduction of the entire process: this role has never been explicitly considered before in the literature of the exemplar-based inpainting-by-patch procedures.

Following in the direction of this reduction, we formalized a new priority index, giving an applicative explanation of the roles played by each one of its terms: the inpainting energy $E_T$, the trustability $C$, and the finite differences $D_M$. We proved that all of them concur together to the uncertainty reduction.

In addition, we proposed and implemented a new dynamic for the update of the priority, with the introduction of the completely new concept of the error redistribution among neighbor points.

The numerous numerical results, both on synthetic and on natural images, have confirmed the expectations: in terms of the mathematical model, in terms of the values we calculated, and in terms of the parameters in the applications. Again, we remark that our method is not parameter-dependent, the values of the parameters having been explicitly determined exploiting uncertainty assumptions.

Our work shows that a deeper understanding of the exemplar-based inpainting-by-patch procedure effectively brings to a reduction in the needed information to achieve correct results, compared with the algorithms of the same class.

Thanks to our new model, we proved a meaningful reduction of the number of neighbor points needed for the reconstruction, when compared with some well known state of the art algorithms using the same inpainting approach.

Our new proposed method reduces the uncertainty in the inpainting process, boosting the amount of information that can be inferred from the training set.

We obtained better results (also reducing the training set) than the ones achieved merely increasing the size of the neighborhood in the direction of a more efficient use of the known information hidden in what we referred to as the *structure* of the given image.

Generally, it is not easy to a priori identify cases of failure of our technique. Its mathematical formulation proves that the number of plausible values to be used for the inpainting is lower than the formulation without the structural term. This aspect confers a lower error probability on our approach. Moreover, the results we reported in this work have been achieved on images with high signal to noise ratio, such that the effects of noise have been reduced, in this first stage.

Moreover, another crucial point is represented by the execution time: the implementation, e.g., of a PatchMatch-like approach [21] can be the focus for future works.

Finally, given our probabilistic interpretation, it is possible to find contact points between the inpainting-by-patch algorithms and NNs. It is well known how NNs are efficient statistical classifiers. Among them, U-nets are capable to inpaint images only using the known part of the given single image itself. Then, this kind of nets uses a training set that is the same of our $T_S$. Initialising a U-net generally requires starting random values, the same way inpainting-by-patch methods do. Then, the major difference between NNs and inpainting-by-patch methods resides in the extremely reduced execution time of NNs, hidden in the NNs interpretability, probably the major open problem in the NNs research field nowadays.

**Author Contributions:** Conceptualization, M.S. and S.R.B.; methodology, M.S.; software, M.S.; validation, M.S.; formal analysis, M.S.; investigation, M.S.; resources, M.S.; data curation, M.S.; writing—original draft preparation, M.S.; writing—review and editing, S.R.B.; supervision, S.R.B. All authors have read and agreed to the published version of the manuscript.

**Funding:** This research received no external funding.

**Institutional Review Board Statement:** Not applicable.

**Informed Consent Statement:** Not applicable.

**Data Availability Statement:** Not applicable.

**Conflicts of Interest:** The authors declare no conflict of interest.

## Appendix A. Numerical Implementation

To calculate $E_{S_T}$ in a suitable algorithmic form, some considerations are useful to be made. In the discrete one-dimensional case, given a row $r$ of the image $I$, we can rewrite $I(r, i + k), r, k, i \in \mathbb{N}$ as a function of past samples using the following equation (In place of a horizontal direction $r$ (i.e., a row), it is possible to consider whatever orientation at

the price to introduce some interpolation methods to compute the missing values on the discrete grid.): $I(r, i + k) = I(r, i) + \Delta^{(1)} I(r, i) + \Delta^{(1)} I(r, i + 1) + ... + \Delta^{(1)} I(r, i + k - 1)$.

**Proof.** From the definition of $\Delta^{(1)} I(r, i)$, given a generic index $i$, the equality $I(r, i) + \Delta^{(1)} I(r, i) = I(r, i + 1)$ stands. Then

$$I(r, i) + \Delta^{(1)} I(r, i) + \Delta^{(1)} I(r, i + 1) + ... + \Delta^{(1)} I(r, i + k - 1) =$$

$$I(r, i + 1) + \Delta^{(1)} I(r, i + 1) + ... + \Delta^{(1)} I(r, i + k - 1) = .... = I(r, i + k),$$

the last equality stands iterating the procedure.  □

Moreover, in general, it is possible to write

$$\Delta^{(1)} I(r, i) = I(r, i + 1) - I(r, i)$$

$$\Delta^{(2)} I(r, i) = [I(r, i + 2) - I(r, i)] - [2\Delta^{(1)} I(r, i)]$$

...

$$\Delta^{(k)} I(r, i) = [I(r, i + k) - I(r.i)] - [a_{k,1} \Delta^{(1)} I(r, i) + ... + a_{k,j} \Delta^{(j)} I(r, i) + ... +$$

$$+ a_{k,k-1} \Delta^{(k-1)} I(r, i)]$$

with the coefficients $a_{k,j} = \binom{k}{j}$, as they come form the Pascal's triangle formulation.

Given a direction, the above expression allows the calculation of the finite difference of order $k$ in $(r, i)$ as a linear combination of pointwise differences (i.e., $I(r, i + 1) - I(r, i)$, $I(r, i + 2) - I(r, i)$,...). In the two-dimensional case, this results in the combination of differences between the central point and a square shaped ring, depending on the maximum finite difference order chosen (see Figure A1). In particular, this generally requires some interpolation or quasi interpolation techniques to provide values along those directions that are not explicitly available.

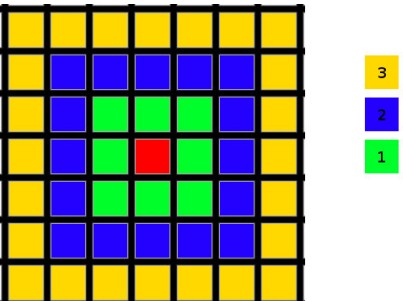

**Figure A1.** An example of points involved in the calculation of finite differences: the red point of coordinates $(i, j)$ is the point to reconstruct; the green points identify the values needed for the calculation of the finite differences of order 1 with respect to $(i, j)$; the blue points, together with the green points, identify the values needed for the calculation of the finite differences of order 2 with respect to $(i, j)$; the yellow points, together with the blue and green points, identify the values needed for the calculation of the finite differences of order 3 with respect to $(i, j)$.

Calculating high order finite differences, in a two-dimensional setting, is time-consuming. Moreover, recalling that the quality of the reconstruction is evaluated by simple observation, we stopped at the first-order finite differences. In addition, to justify this choice also from a perceptive point of view, it is quite evident that a human observer can easily determine the class of continuity of a function only until the first order (see Figure A2). The same reasons have been used to justify the definition of the data term in $P^*(i, j)$.

In the applications, it happens to have different candidate values with the same $E_T$: when this is the case, we choose their mean value as the best match. In Algorithm A1 we show the pseudo-code of the whole method theoretically described in the previous sections.

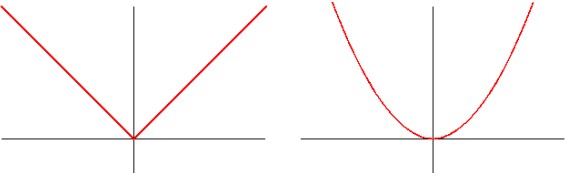

**Figure A2.** A $C^0$ function on the left and a $C^k$ function on the right: a human observer can easily perceive the continuity of the left function but cannot determine $k$ of the right one.

---

**Algorithm A1** Inpaint by Finite Differences

---

**procedure** INPAINT(I,M,$T_S$,$\chi_S$,n)
    ▷ I=Image    ▷ M=To-Be-Inpainted area    ▷ $T_S$=Training Set    ▷ $\chi_S$=support of Chi    ▷ n= order of finite differences

    P=M

    $E_T \leftarrow EnergyCalculation(I, M, T_S, \chi_S, n)$
                                           ▷ first calculation of $E_T$
*loop* :
    **for** $(ii, jj) \in M$ **do**
        $P \leftarrow updatePriority(I, M, T_S, n, E_T, P)$
                                         ▷ return the priority for each point to inpaint

        $I(find(P == max(P))) \leftarrow T_S(bestFit(find(P == max(P))))$
                                         ▷ inpaint the point with highest priority

        $E_T \leftarrow EnergyCalculation(I, M, T_S, \chi_S, n)$
                                         ▷ update $E_T$ after the inpainting
    *goto loop* :

**procedure** $E_T \leftarrow$ ENERGYCALCULATION(I,M,$T_S$,$\chi_S$,n)
    ▷ I=Image    ▷ M=To-Be-Inpainted area    ▷ $T_S$=Training Set    ▷ $\chi_S$=support of Chi    ▷ n= order of finite differences

    bestFit=I

    **for** $(ii, jj) \in M$ **do**
        $E_{T_{min}} = 256 \cdot ones(size(M))$

        **for** $(i, j) \in T_S$ **do**
            $E_C = sum(abs(I(ii \in \chi_S, jj \in \chi_S) - T_S(i \in \chi_S, j \in \chi_S)))$
            $E_{C_{norm}} = Norm(E_C)$
                                   ▷ normalization of $E_C$ such that $0 \leq E_C \leq 1$

            $E_S = 0$
            **for** d=1:n **do**
                **for** k=1:8*k **do**
                     $E_S = E_S + sum(abs(\Delta(I(ii \in \chi_S, jj \in \chi_S), k, d) - \Delta(T_S(i \in \chi_S, j \, in\chi_S), k, d)))$
            $E_{S_{norm}} = Norm(E_S)$
                                   ▷ normalization of $E_S$ such that $0 \leq E_S \leq 1$

            $E_{T_*} = E_{C_{norm}} + E_{S_{norm}}$
            **if** $E_T < E_{T_{min}}$ **then**
                $bestFit(ii, jj) \leftarrow T_S(i, j)$
                $E_T(ii, jj) \leftarrow E_{T_*}$

**procedure** $P \leftarrow$ UPDATEPRIORITY(I,M,$T_S$,$\chi_S$,n,$E_T$,P)
    ▷ I=Image  ▷ M=To-Be-Inpainted area  ▷ $T_S$=Training Set  ▷ $\chi_S$=support of Chi  ▷ n= order of finite differences  ▷ $E_T$= Inpainting Energy                                         ▷ P= Priority

    **for** $(ii, jj) \in M$ **do**
        $C(ii, jj) \leftarrow calculateC(I, M, \chi_S)$
        $D(ii, jj, \theta)) \leftarrow calculateD(I, M, \chi_S, n)$
        $P(ii, jj) \leftarrow (C(ii, jj) + max_\theta D(ii, jj, \theta)) \cdot E_T(ii, jj)$
        ▷ C is the trustability, $max_\theta D$ is the maximum of the finite differences; their calculation details are omitted for brevity

---

The color images have been managed operating separately on the RGB channels and joined at the end of the procedure. No blending of any sort (e.g., Poisson blending) has been used to adjust the final results.

**Abbreviations**

| | |
|---|---|
| $a$ | row shift value in supp $\chi$; |
| $a^*$ | normalization coefficient for $P^*$; |
| $\alpha$ | normalization coefficient for $D$; |
| $b$ | column shift value of supp $\chi$; |
| $b^*$ | normalization coefficient for $P^*$; |
| $\beta_k$ | normalization coefficient for $E_{S_T}$; |
| $C$ | trustability or confidence term; |
| $\chi$ | neighborhood function; |
| $D$ | state of the art data term; |
| $D_M$ | proposed new data term; |
| $d$ | pointwise distance between two pixels in $I$; |
| $d_{CNorm}$ | normalized pointwise content-related distance between two pixels in $I$; |
| $d_{SNorm}$ | normalized pointwise structure-related distance between two pixels in $I$; |
| $\Delta_\theta^{(k)}$ | finite difference operator of order $k$ in direction $\theta$; |
| $\partial\Omega$ | boundary of $\Omega$; |
| $E_C$ | single point, content-related, inpainting energy; |
| $E_{C_T}$ | total content-related inpainting energy; |
| $E_I$ | inpainting energy; |
| $E_{M_C}$ | total patch content-related probability of a match in $T_S$ given supp $\chi$; |
| $E_{M_S}$ | total patch structure-related probability of a match in $T_S$ given supp $\chi$; |
| $E_{P_C}$ | pointwise content-related probability of a match in $T_S$ given supp $\chi$; |
| $E_{P_S}$ | pointwise structure-related probability of a match in $T_S$ given supp $\chi$; |
| $E_S$ | single point, structure-related, inpainting energy; |
| $E_{S_T}$ | total structure-related inpainting energy; |
| $E_T$ | total inpainting energy; |
| $I$ | image under investigation; |
| $i, h, r, p$ | row coordinate of a pixel in $I$; |
| $j, m, s, q$ | column coordinate of a pixel in $I$; |
| $k$ | generic order of the finite differences; |
| $l$ | maximum order of the finite differences; |
| $\ell$ | discrete Lebesgue space; |
| $\Lambda$ | summation domain; |
| $M$ | number of rows in $I$; |
| $N$ | number of columns in $I$; |
| $n$ | unitary vector orthogonal to $\partial\Omega$; |
| $\nabla^\perp$ | isophote vector; |
| $\Omega$ | to-be-inpainted area; |
| $P$ | state of the art scanning priority index; |
| $P^*$ | proposed scanning priority index; |
| $R_k$ | direction-related normalization coefficient for $E_{S_T}$; |
| $\mathcal{S}$ | specific configuration in $S_C$; |
| $S_C$ | set of equivalence classes; |
| supp $\chi$ | support of the neighborhood function; |
| $t$ | time; |
| $T_S$ | training set; |
| $\theta$ | specific orientation of the finite difference; |
| $\Theta_k$ | set of possible and valid orientations $\theta$ given the order $k$; |
| $W$ | bit depth; |
| $U$ | uncertainty. |

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
