# Peer review of "Inpainting in Discrete Sobolev Spaces: Structural Information for Uncertainty Reduction"

_applsci, doi:10.3390/app13169405_

Round 1

Reviewer 1 Report

The paper presents an exemplar-based inpainting method to restore missing regions in an image. The main contribution is a new mathematical function that incorporates both the content and structure of the image. The authors also introduce a new priority index that determines the order of filling the missing points. The proposed method is novel and theoretically sound, as it uses finite difference terms to approximate the derivatives in Sobolev spaces. The work was previously presented in a preprint server as arXiv:2211.03711v1.

The paper is well written, sophisticated and of high quality. It demonstrates the novelty and computational efficiency of the proposed method.

However, I have some minor comments that could improve the paper:

  1. The paper lacks a thorough analysis of the failure cases of the proposed method. The authors only show some examples where their method performs better than other patch based methods, but not discuss the limitations and challenges of their method compared to limitations of others. I think this is important to provide a fair and comprehensive evaluation of the proposed method, and to identify possible directions for future work.
  2. The authors claim that their work can help to develop more interpretable neural network based methods, which are considered superior in inpainting. However, they do not explain or speculate how this can be achieved, given the different nature of the two approaches. I would like to see some discussion on this point, or some references to existing works that attempt to bridge the gap between exemplar-based and neural network based methods.

Overall, I think the paper is well done and deserves publication in the journal.

Author Response

Dear Reviewer,

Thanks for your precious comments and your time. In what follows, you can find the list of all the corrections we performed (where we attach also your questions), based on your suggestions. We hope they can be considered appropriate. When possible, we have included all the corrections in the paper; where not, we have provided a direct answer.

The structure of the work has been maintained but some parts have been added.

In what follows we refer to the sections where the modifications have been done.

-------------------------------------------------------

1 The paper lacks a thorough analysis of the failure cases of the proposed method. The authors only show some examples where their method performs better than other patch based methods, but not discuss the limitations and challenges of their method compared to limitations of others. I think this is important to provide a fair and comprehensive evaluation of the proposed method, and to identify possible directions for future work.

In the conclusion section we have added a some sentences to better explain the advantage of our method compared with what available in the literature.

There are many inpainting techniques available and an exhaustive comparison would be really difficult, considering also that even very well known algorithms, like the Criminisi one, do not have a public official available code.

We have not experimented evident cases of failure but we have also to say that the evaluation of the results is left to the subjectivity of the human observer.

Generally, exploiting the mathematical definition and the cardinality of the equivalence classes introduced, the structural term considerably reduces the number of plausible values to inpaint, fact that decreases (or in some cases eliminates) any errors. This is particular evident in the convergence to the original image, when the to-be-inpainted area is drawn over a known image. A particular difficult problem is to inpaint an image where the most of the content is totally missing (e.g., like drawing a missing painting when only the frame is known). Even if it is not possible, for us, to determine a priori failure cases, what we can say is that mathematically (i.e., theoretically) our choice will always be better (or at least equal) than the one not introducing the structural term. A drawback of the method is its execution time, if compared to the Criminisi-like approaches.

To explain what stated in the previous sentences we added, in the conclusion section, from line 522 to line 529, the following text:

" Generally, it is not easy to a priori identify cases of failure of our technique. Its mathematical formulation proves that the number of plausible values to be used for the inpainting is lower than the formulation without the structural term. This aspect confers a lower error probability on our approach. Moreover, the results we reported in this work have been achieved on images with high signal to noise ratio, such that the effects of noise have been reduced, in this first stage.

Moreover, another crucial point is represented by the execution time: the implementation, e.g., of a PatchMatch-like approach [4] can be the focus for future works."

  1. The authors claim that their work can help to develop more interpretable neural network based methods, which are considered superior in inpainting. However, they do not explain or speculate how this can be achieved, given the different nature of the two approaches. I would like to see some discussion on this point, or some references to existing works that attempt to bridge the gap between exemplar-based and neural network based methods.

We thank the reviewer for this really interesting observation. To the best of our knowledge there are not contributions in the literature investigating the connection between the inpainting-by-patch algorithm and NNs. Moreover, the investigation of this connection can be the topic for new works on which we are currently involved. What we can say is that the inpainting-by-patch class of algorithms looks really similar to U-nets, where the network is trained only on the known part of the image itself. This known part represents the training set of the network. Moreover, NN provides probabilities in output, in a similar manner to what happens given our probabilistic interpretation. It is also well known that NN are statistical classifiers: their power stands in the reduced execution time needed for the classification. This last aspect. i.e., the interpretability of NNs,  still represents an open topic, probably the most important one, in the NNs research. At the moment we have not further details concerning this similarity.

To explain what stated in the previous sentences we added, in the conclusion section, from line 530 to line 538, the following text:

“ Finally, given our probabilistic interpretation, it is possible to find contact points between the inpainting-by-patch algorithms and NNs. It is well known how NNs are efficient statistical classifiers. Among them, U-nets are capable to inpaint images only using the known part of the given single image itself. Then, this kind of nets uses a training set that is the same of our TS . Initialising a U-net generally requires starting random values, the same way inpainting-by-patch methods do. Then, the major difference between NNs and inpainting-by-patch methods resides in the extremely reduced execution time of NNs, hidden in the NNs interpretability, probably the major open problem in the NNs research field nowadays.”

Reviewer 2 Report

The manuscript presents a mathematical model to investigate the inpainting problem in Sobolev spaces, an interesting research topic. Although experiments and results are interesting, some major observations must be addressed:

1) Figures 4, 9, 10, and 18 have low image quality for line style, colors, and font size. There is no name for the axes in the graphs. The conditions of these graphs detract from the work because it is impossible to appreciate the differences or interpret the results.

2) The authors introduced two hypotheses:

    a) the system satisfies the superposition effects principle (linearity);

    b) the system is Markovian [25].

    However, these are affirmations or, in any case, assumptions. A hypothesis must be falsifiable. That is, there must be the possibility of giving a counterexample to refute the hypothesis. Please correct it or explain why the statements made are hypotheses.

3) The authors mix the terms formulae with equations. Please, unify the term.

Author Response

Dear Reviewer,

Thanks for your precious comments and your time. In what follows, you can find the list of all the corrections we performed (where we attach also your questions), based on your suggestions. We hope they can be considered appropriate. When possible, we have included all the corrections in the paper; where not, we have provided a direct answer.

The structure of the work has been maintained but some parts have been added.

In what follows we refer to the sections and lines where the modifications have been done.

The manuscript presents a mathematical model to investigate the inpainting problem in Sobolev spaces, an interesting research topic. Although experiments and results are interesting, some major observations must be addressed:

1) Figures 4, 9, 10, and 18 have low image quality for line style, colors, and font size. There is no name for the axes in the graphs. The conditions of these graphs detract from the work because it is impossible to appreciate the differences or interpret the results.

 Fig. s 4, 9, 10, 18 have been corrected and made clearer.

2) The authors introduced two hypotheses:

  1.   a) the system satisfies the superposition effects principle (linearity);
  2.   b) the system is Markovian [25].

    However, these are affirmations or, in any case, assumptions. A hypothesis must be falsifiable. That is, there must be the possibility of giving a counterexample to refute the hypothesis. Please correct it or explain why the statements made are hypotheses.

Between line 364 and line 377 we have corrected and introduced some more sentences to explain the need for linearity and Markovianity. In particular, we have replaced the word hypotheses with the word assumptions, to put in evidence that our choice is, indeed, needed only to make the problem solvable.

"We can figure C to be as a field (e.g., of force) where all the points are interconnected: a perturbation occurring somewhere would cause the modification of the entire configuration. To make this update numerically plausible we postulate two assumptions: we assume

the system must satisfy the superposition effects principle (linearity);

the system must be Markovian .

Both these assumptions are needed with the uinque scope to make the problem computable. In fact, the first one allows to update C(i,j) in I/Omega summing the effects separately; the second one limits the radius of influences of the perturbation to the bordering neighborhoods. If the linearity is missing, the computation would not arrive to an end, because changing the value of one pixel will influence the minima connected with the other points in Omega. If the the system is not Markovian, the changing in the value of a missing point would propagate all over the other values, producing an infinite loop, as in the case of the lack of linearity."

3) The authors mix the terms formulae with equations. Please, unify the term.

At line 140 we replaced the word "formula" with the word "equation", to unify the language.
